# Antagonistic odor interactions in olfactory sensory neurons are widespread in freely breathing mice

Joseph D. Zak [1,2 ✉], Gautam Reddy[3], Massimo Vergassola [4] & Venkatesh N. Murthy [1,2 ✉]

Odor landscapes contain complex blends of molecules that each activate unique, overlapping populations of olfactory sensory neurons (OSNs). Despite the presence of hundreds of OSN subtypes in many animals, the overlapping nature of odor inputs may lead to saturation of neural responses at the early stages of stimulus encoding. Information loss due to saturation could be mitigated by normalizing mechanisms such as antagonism at the level of receptor-ligand interactions, whose existence and prevalence remains uncertain. By imaging OSN axon terminals in olfactory bulb glomeruli as well as OSN cell bodies within the olfactory epithelium in freely breathing mice, we find widespread antagonistic interactions in binary odor mixtures. In complex mixtures of up to 12 odorants, antagonistic interactions are stronger and more prevalent with increasing mixture complexity. Therefore, antagonism is a common feature of odor mixture encoding in OSNs and helps in normalizing activity to reduce saturation and increase information transfer.

[1] Department of Molecular & Cellular Biology, Harvard University, Cambridge, MA 02138, USA. [2] Center for Brain Science, Harvard University, Cambridge, MA 02138, USA. [3] NSF-Simons Center for Mathematical & Statistical Analysis of Biology, Harvard University, Cambridge, MA 02138, USA. [4] Laboratoire de Physique de l'Ecole Normale Supérieure, ENS, Université PSL, CNRS, Sorbonne Université, Université de Paris, Paris F-75005, France. ✉email: jzak@fas.harvard.edu; vnmurthy@fas.harvard.edu

The number of distinct types of sensory neurons is usually far smaller than the number of distinct stimuli that an animal needs to detect, necessitating individual neurons to be receptive to multiple stimuli. This feature is especially relevant in the olfactory system where the number of odorous molecules vastly exceeds the repertoire of receptor types[1–3]. Further, the olfactory world at any instant consists of complex mixtures of odorants, with individual olfactory receptors encountering multiple odorants at once (Fig. 1a). Therefore, how multiple ligands interact at single odorant receptors will define the mode of coding in, and the capacity of, the olfactory system. A large fraction of the olfactory sensory neurons (OSNs) in mammals express a family of odorant receptors that are G-protein coupled receptors[4–6]. If the ligand-receptor binding kinetics are modeled with a single affinity parameter, simultaneous activation by multiple ligands will be simply determined by the relative affinities, until saturation[7–10]. However, it is becoming increasingly clear that more complex interactions can occur.

Recent theoretical work has shown that a variety of nonlinear interactions among multiple ligands at the same receptor can readily arise with a simple two-step model of receptor activation[11] (Fig. 1b, c). Experimental work in vitro in OSNs has suggested the existence of nonlinear interactions, especially antagonism or partial antagonism[12–20]. However, the prevalence of these interactions, especially in living animals within the constraints of natural sniffing dynamics, has not been explored. At a more basic level, evidence for multistep receptor activation has also been sparse. For example, odorants with different affinities for a given receptor could also have distinct maximal activation if their efficacies are different[17], but few studies have systematically explored this aspect.

Elucidating the principles of mixture interactions in OSNs is important for understanding odor coding. Odor identity and abundance are widely accepted to be represented by a combination of OSNs[21–23]. While each odorant activates a discrete pattern of sensory inputs, there can be considerable overlap in patterns of OSN activation corresponding to different odorants[24–27]. Since naturalistic odor stimuli are complex blends of many odorants, even relatively simple odor blends may saturate a large fraction of the complement of OSNs, thereby limiting their information coding capacity. More generally, even before saturation sets in, nonlinear interactions among multiple odorants in a given OSN may pose challenges for downstream decoding of odor identity[28,29].

We systematically characterize mixture interactions at OSNs in their native environment using two-photon imaging of calcium responses to odor stimuli. Imaging populations of OSN axon terminals in the olfactory bulb glomerular layer affords excellent signal-to-noise ratio and direct access to information conveyed to the brain. We also adapted a method to chronically image OSN somata in the olfactory epithelium of freely breathing mice, which allows us to bypass any influence of top-down neural feedback and to directly access the result of odor transduction. By delivering odors individually and in binary mixtures at concentrations that varied over three orders of magnitude, we uncover evidence for multistep activation of ORs and for widespread antagonistic interactions. We also find such antagonistic interactions in responses to complex mixtures containing up to 12 components in single OSNs. Our data strongly support a role for mixture suppression of OSN activity as a normalizing mechanism in olfactory stimulus encoding, which could enhance odor information transfer.

## Results

**Antagonism measured in the glomerular layer of the OB.** Odor responses in OSNs can be measured with excellent sensitivity in the glomerular layer of the olfactory bulb, where axons of a given receptor type converge, allowing signal averaging. We used *OMP-GCaMP3* mice and imaged axonal calcium responses through a cranial window over the dorsal surface of both olfactory bulbs. For two odorants, Methyl tiglate and Isobutyl propionate (Fig. 2a; Table 1), we measured odor responses in glomeruli across a range

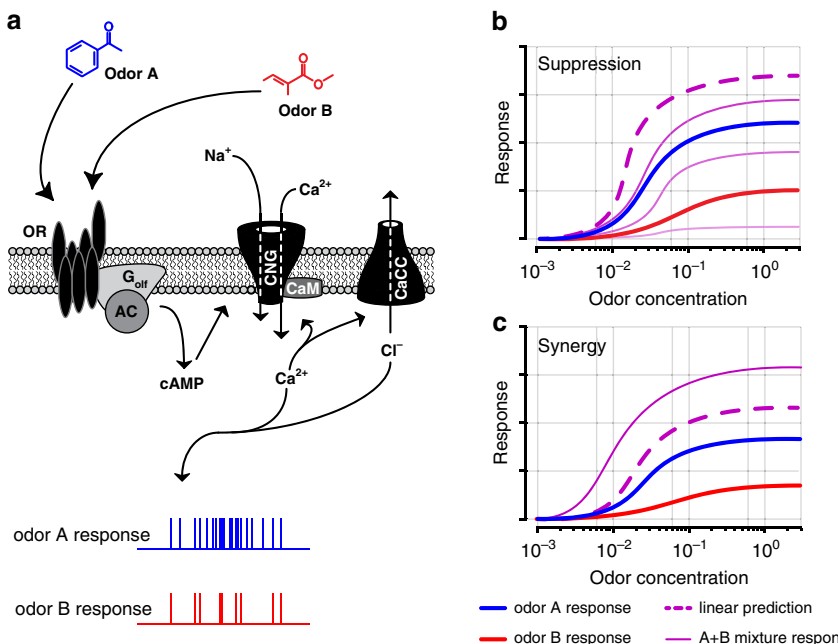

**Fig. 1 Schematic of signal transduction and nonlinear odor mixture interactions in OSNs. a** Schematic of sensory transduction in OSNs. Odors generate responses in OSNs with different levels of efficacy. Olfactory receptor (OR), adenylyl cyclase (AC), cyclic nucleotide gated channel (CNG), calmodulin (CaM), calcium-activated-chloride channel (CaCC). **b** Mixtures of odors can give rise to nonlinear responses in OSNs. Three solid purple lines show hypothetical mixture suppression (antagonism) at varying levels with respect to the component-based linear prediction (dashed purple). **c** Although less common, mixture responses can also display synergy where the mixture response exceeds the component-based linear prediction.

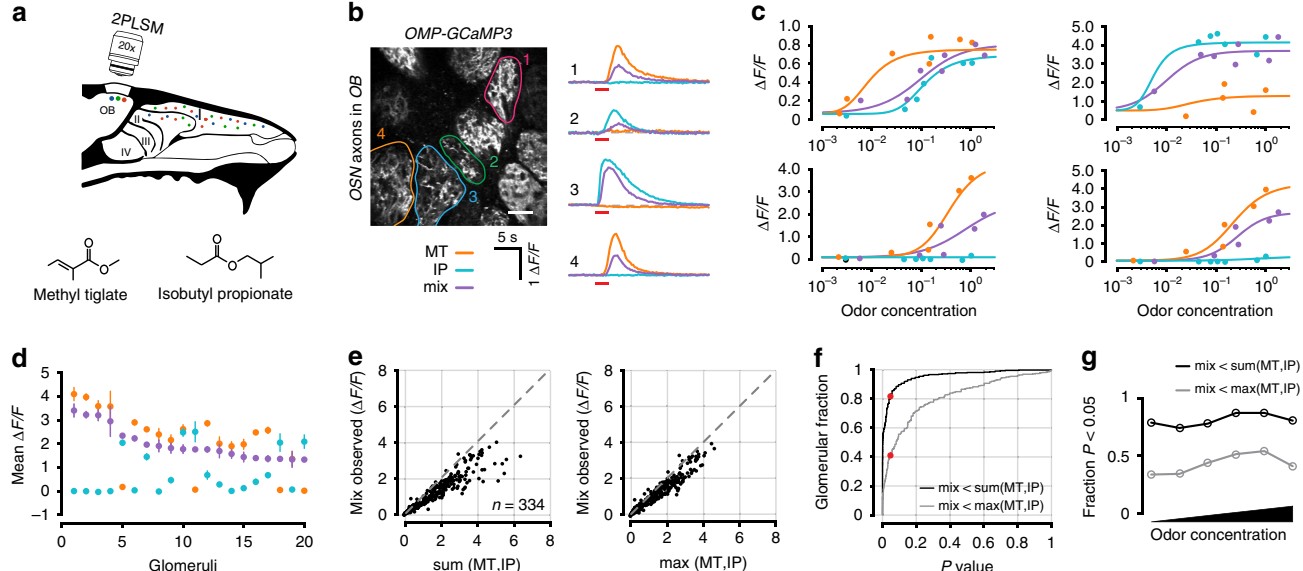

**Fig. 2 Optical measurements of antagonism in OSN axon terminals. a** Experimental setup. Odor responses in OSN axon terminals were measured at olfactory bulb glomeruli through a cranial window. **b** Example image of glomeruli and selected ROIs from one of nine mice in this dataset. Responses for two odors (Methyl tiglate and Isobutyl propionate) and their mixture from selected glomeruli are shown on right as ΔF/F time courses. Colored circles correspond to ROIs in image. Red bar under traces denotes odor delivery time. Scale bar in image is 50 μm. **c** Example dose response curves from four selected glomeruli. Each point is the average of 3–5 trials. Odor concentrations are normalized to the largest measured concentration (see Methods). For each point, odor mixtures are the summation of both individual odors. **d** Data from 20 randomly selected glomeruli. Each point is the mean of the three largest responses from each odor. Error bars are SEM. **e** Left, comparison of the observed mixture response against the linear sum of both mixture components or (right) against the maximum response generated by either mixture component. **f** Cumulative distribution of rank-sum test P values obtained for 334 glomeruli for linear sum comparison (black) and comparison to maximum component response (grey). Red dots mark P values < 0.05. **g** Fraction of glomeruli that showed significant antagonism in each of six concentration bins. Each bin contains three adjacent concentrations.

**Table 1 Odor information related to Figs. 2, 4 and 6.**

| Name/Formula | Structure | CAS number | Molecular Weight (g/mol) | Standard Vapor Pressure (mmHg at 25°C) |
|---|---|---|---|---|
| Methyl tiglate $C_6H_{10}O_2$ | | 6622-76-0 | 114.14 | 13.3790 |
| Isobutyl propionate $C_7H_{14}O_2$ | | 540-42-1 | 130.18 | 6.4700 |

of concentrations spanning three orders of magnitude. We then made a binary mixture of the two odorants and measured OSN responses at the same glomeruli (Fig. 2b, c). When plotting the responses as a function of concentration, the mixture concentration at each point represents the sum of the two individual odors—this applies to all experiments where concentration series are used. From nine mice, we identified 334 glomeruli that responded to either of the odors, or the mixture of the two. For this odor pair, many glomeruli reached different saturation levels for the same odor (Fig. 2c).

To estimate antagonism at each glomerulus, we first calculated the average response to the three highest odor concentrations of each individual odor (Fig. 2c, d). We then tested whether the observed mixture responses for the same three concentrations were significantly below the linear sum of the response to the mixture components (rank-sum test of trial replicates). Significant mixture suppression was found in 80.5% (277/344) of glomeruli that responded to at least one odor (Fig. 2e, f).

There are, however, two important considerations in interpreting our result. First, it remains possible that the largest calcium signals observed for individual odors may represent the

average maximum physiologically-bounded firing rate across all OSNs of a common type. Similarly, the largest signals we observed for individual odors may be bounded by calcium indicator saturation. Given these constraints, it is plausible that the observed mixture responses can never reach the linear sum of the two mixture components and are simply bounded by the largest response observed for either component alone. To account for this possibility, we compared the observed mixture responses and maximum response for either odor. When using this more conservative metric, we found evidence for antagonism in 41.7% (144/344) of the glomeruli (Fig. 2e-f). These data indicate that, even by a conservative measure, antagonism is widespread in OSNs. Lastly, to ensure that our observations were not only relevant to the highest odor concentrations, we measured the fraction of antagonistic interactions on a sliding scale (Fig. 2g). We found that regardless of odor concentration, antagonism was present in a constant fraction of glomeruli. These data cannot be easily explained by one-step competitive binding models, and provide further evidence that odorant- binding to and activation of odor receptors may be decoupled, as predicted by our earlier theoretical work[11].

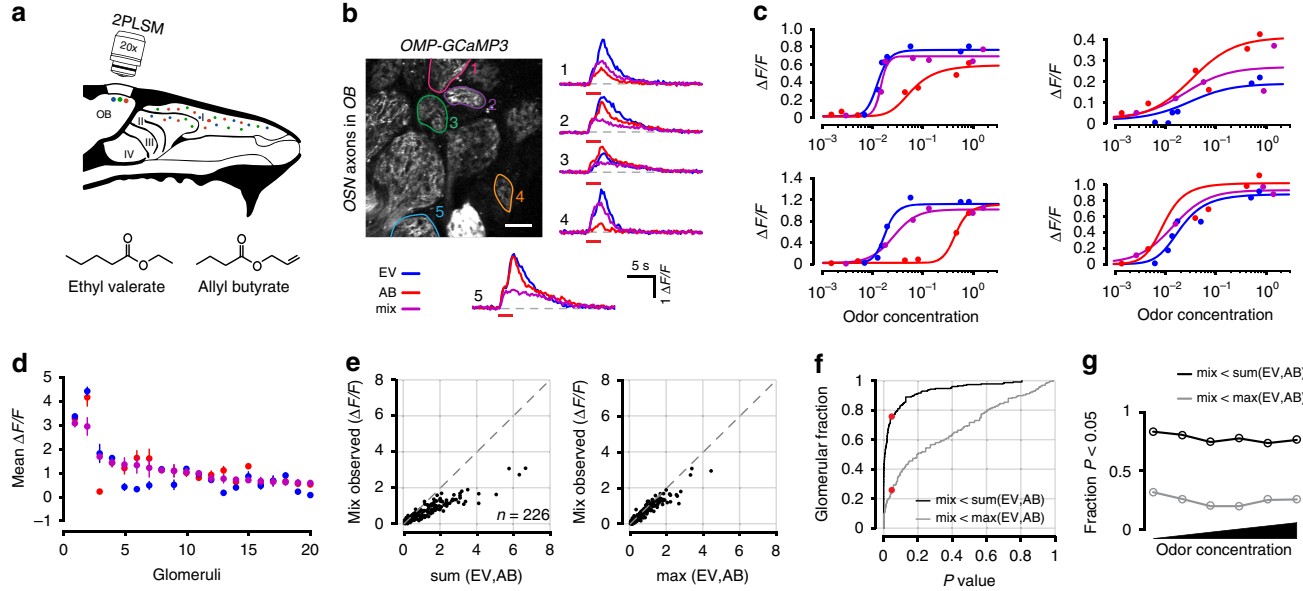

**Fig. 3 Antagonism in OSN axon terminals for odors with overlapping activation patterns. a** Experimental setup. Same as Fig. 1, but for an odor pair with overlapping response profiles (Ethyl valerate and Allyl butyrate). **b** Image of glomeruli and selected ROIs from one of five mice in this dataset. Odor responses for each odor and their mixture are shown on the right. Colored circles correspond to ROIs in image. Red bar under traces denotes odor delivery time. Scale bar in image is 50 µm. **c** Example response curves from selected glomeruli. Data are plotted the same as in Fig. 1. **d** Data from 20 randomly selected glomeruli, rank ordered by the mixture response. Each point is the mean response to the three highest concentrations of each odor. Error bars are SEM. **e**–**g** Data from 226 glomeruli, plotted the same as in Fig. 1.

**Table 2 Odor information related to Figs. 3 and 4.**

| Name/Formula | Structure | CAS number | Molecular Weight (g/mol) | Standard Vapor Pressure (mmHg at 25°C) |
|---|---|---|---|---|
| Ethyl valerate $C_7H_{12}O_2$ | | 539-82-2 | 130.18 | 4.7450 |
| Allyl butyrate $C_7H_{12}O_2$ | | 2051-78-7 | 128.17 | 4.4430 |

In these experiments, glomerular responses were highly non-overlapping for the odor pair tested. To ensure that antagonism did not arise from unique interactions between these two odors, and to demonstrate generalization of the phenomenon with other odor pairs, we repeated our experiments using another odor pair, Ethyl valerate and Allyl butyrate (Fig. 3a; Table 2). This pair was specifically selected because of their highly overlapping OSN activation (Fig. 2b). Mixture interactions for this pair of odors were in close agreement with those estimated above for the other odor pair. Using the same approach that we previously described, from 226 glomeruli in five mice, we found antagonism in 76.6% (173/226) of glomeruli when comparing the summed responses and 25.7% (58/226) when comparing against the maximum (Fig. 3e, f). Again, the fraction of glomeruli that exhibited mixture suppression was concentration invariant (Fig. 3g).

Across all glomeruli from both odor pairs where saturating responses were observed for an odor, we could then fit response curves using Eq. (1) (see Methods). We used stringent criteria that ensured only well-activated glomeruli were included by imposing thresholds on the mean-squared error from the best fit parameters. When comparing the fits from glomerular responses to the odors used in Fig. 2 (Fig. 4a; also see Table 1), the distribution and mean Hill coefficients were similar ($2.67 \pm 0.16$, $n = 48$ for Methyl tiglate and $2.53 \pm 0.32$, $n = 22$ for Isobutyl propionate; $P > 0.05$, Rank-sum test; Fig. 4b). Early experiments

with synthetic calcium indicators or intrinsic imaging of glomeruli estimated Hill coefficients to be close to 1, but more recent experiments appear to measure higher values[9,10]. We then compared the Hill coefficients obtained from the curves fit to the mixture responses (mean $= 2.21 \pm 0.16$, $n = 30$; Fig. 4d) with those from each of the components alone. We found no differences in any of the comparisons ($P > 0.05$, Rank-sum test).

When we fit the glomerular responses to the odor pair used in Fig. 3 (Fig. 4c; also see Table 2), the distribution and the means of the Hill coefficients were remarkably similar ($2.36 \pm 0.19$, $n = 44$ for Ethyl valerate and $2.29 \pm 0.21$, $n = 39$ for Allyl butyrate; $P > 0.05$, Rank-sum test; Fig. 4d), and comparable to those for the odor pair discussed above. We again compared the Hill coefficients extracted from the data fit to the glomerular mixture responses (mean $= 2.26 \pm 0.22$, $n = 32$; Fig. 4d) to each of the Hill coefficient distributions from the mixture components and found no differences across all comparisons ($P > 0.05$, Rank-sum test).

**Imaging odor responses in individual OSNs.** Our glomerular imaging results suggest that mixture suppression through antagonism is widespread. However, an alternative mechanism that could produce such an effect may operate through GABAb- or D2-mediated suppression of OSN axon terminals[30–33].

To avoid circuit interactions between different OSNs, we directly imaged OSN somata in situ in the olfactory epithelium[34],

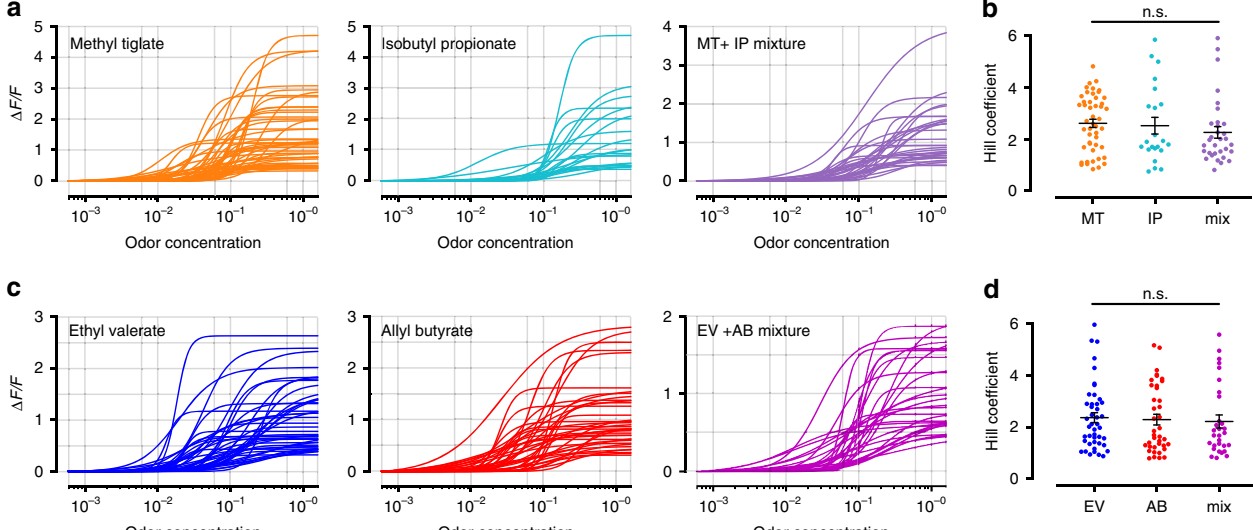

**Fig. 4 Hill coefficients of odor responses in OSN axon terminals. a** Example response curve fits from individual OSNs for odors in Table 1 and their mixture. **b** Hill coefficient distributions from the individual OSNs plotted in part (**a**) ($n = 48$ for Methyl tiglate, $n = 22$ for Isobutyl propionate, $n = 32$ for MT + IP mixture; $P > 0.05$ for all comparisons; two-sided rank-sum test). Data are presented as mean ± SEM **c** Example response curve fits from individual OSNs for odors in Table 2 and their mixture. **d** Hill coefficient distributions from the individual OSNs plotted in part (**c**) ($n = 44$ for Ethyl valerate, $n = 39$ for Allyl butyrate, $n = 30$ for EV + AB mixture; $P > 0.05$ for all comparisons; two-sided rank-sum test). Data are presented as mean ± SEM.

where there is no evidence for efferent modulation that is directly linked to olfactory circuitry[35]. We first characterized the odor tuning and response dynamics of individual cells to determine whether the response properties of sensory neurons in the dorsal recess (zone 1) of the olfactory epithelium[36,37] are congruent with those observed in glomeruli in the dorsal olfactory bulb, and to ensure that we sample from a heterogeneous population of OSNs. We used a panel of 32 monomolecular odors (see Table 3) that activate glomeruli on the dorsal olfactory bulb across a wide range of densities (Fig. 5a, d, f). We then imaged individual cells in the epithelium (Fig. 5b) and compared their tuning to glomeruli in the olfactory bulb. Across odors, there was a strong relationship between the fraction of activated glomeruli and individual OSNs ($r = 0.750$, $P < 0.0001$, Fig. 5e). We found a similar relationship when we compared the mean response magnitude at both imaging sites across all glomeruli and somata ($r = 0.598$, $P < 0.003$, Fig. 5g). These data provide new functional evidence that support anatomical data for a conserved zonal organization between the olfactory epithelium and the olfactory bulb[37].

We also found that the calcium response kinetics across all OSNs in a field of view are highly diverse, and that the response waveform for individual cells is remarkably stable across trials of the same odor when respiration is stabilized (Supplementary Fig. 1). Together these data suggest that a diverse and heterogeneous array of receptor subtypes are present within a relatively restricted patch of olfactory epithelium.

**Antagonism in individual sensory neurons.** Given that OSNs in the dorsal recess share many of the same odor tuning characteristics as dorsal olfactory bulb glomeruli, we tested whether antagonism could be observed at the single-cell level using the odor pair we used for imaging glomerular responses. We repeated the experiment described in Fig. 2 and collected data from 964 individual sensory neurons using the odor pair, Methyl tiglate and Isobutyl propionate (Table 2). We again detected nonlinear interactions between the odors (Fig. 6b, c). Mixture-suppression was readily observed in individual cells (54.6% (527/964) for sum comparison and 22.0% (212/964) maximum comparison; Fig. 6e,

f). The fraction of cells that showed mixture suppression was smaller than in the glomerular data, which may be attributed to noise from two sources. First, the measured signal at glomeruli represents the population response across all ~10,000 OSNs of a common subtype, compared with single OSN cell bodies in the epithelium imaging. The second source of noise may arise from image acquisition, through the number of pixels contributing to the measured signal. While we typically imaged 10–20 densely packed glomeruli per field of view, in the epithelium, we imaged dozens to hundreds of sparsely arranged cells, thereby substantially reducing the number of pixels contributing to our signal.

To ensure that our observations did not arise from more noisy measurements at the single-cell level, we measured the fraction of mixture responses that exceeded the linear prediction of the mixture components, a phenomenon known as synergy. We first measured the fraction of mixture responses that exceeded maximum of either of the mixture components. We found only 11.8% (114/964) of OSNs exceeded the maximum response of either component (Fig. 6e, f); however, it should be noted that for synergy to occur the measured OSN response should exceed the linear summation of the response of both odor components. In our dataset, we found only 5/964 cells where the mixture response was significantly greater than the sum of the mixture components. Both fractions are well below those we measure for antagonism and consistent with previous reports that antagonism is the predominant mixture interaction when compared with synergy[38].

We also fit the odor responses in individual OSNs and compared the extracted Hill coefficients to those obtained by imaging populations of OSN at their axon terminals (Supplementary Fig. 2). We predicted that the fits for individual OSNs may be sharper, and therefore have larger Hill coefficients than glomerular fits because measurements from glomeruli average hundreds of OSNs that may have heterogenous odor sensitivities. However, when we made these comparisons, we found no difference between individual OSN and glomerular Hill coefficients for the same odors (Methyl tiglate, OSN mean = 3.31 ± 0.76, $n = 6$, glomerular mean = 2.67 ± 0.16, $n = 48$; Isobutyl

**Table 3 Odor information related to Figs. 5 and 7 .**

| Index | Name/Formula | Structure | CAS number | Molecular Weight (g/mol) | Standard Vapor Pressure (mmHg at 25°C) |
|---|---|---|---|---|---|
| 1 | Ethyl tiglate $C_7H_{12}O_2$ | | 5837-78-5 | 128.17 | 4.2690 |
| 2 | Allyl tiglate $C_8H_{12}O_2$ | | 7493-71-2 | 140.18 | 1.2720 |
| 3 | Hexyl tiglate $C_{11}H_{20}O_2$ | | 16930-96-4 | 184.27 | 0.0520 |
| 4 | Methyl tiglate $C_6H_{10}O_2$ | | 6622-76-0 | 114.14 | 13.3790 |
| 5 | Isopropyl tiglate $C_8H_{14}O_2$ | | 1733-25-1 | 142.20 | 1.8770 |
| 6 | Citronellyl tiglate $C_{15}H_{26}O_2$ | | 24717-85-9 | 238.37 | 0.0036 |
| 7 | Benzyl tiglate $C_{12}H_{14}O_2$ | | 37526-88-8 | 190.24 | 0.0010 |
| 8 | Phenylethyl tiglate $C_{13}H_{16}O_2$ | | 55719-85-2 | 204.27 | 0.0010 |
| 9 | Ethyl propionate $C_5H_{10}O_2$ | | 105-37-3 | 102.13 | 35.9000 |
| 10 | 2-Ethyl hexanal $C_8H_{16}O$ | | 123-05-7 | 128.21 | 1.8000 |
| 11 | Propyl acetate $C_5H_{10}O_2$ | | 109-60-4 | 102.13 | 35.2230 |
| 12 | 4-Allyl anisole $C_{10}H_{12}O$ | | 140-67-0 | 148.20 | 0.1650 |
| 13 | Ethyl Valerate $C_7H_{12}O_2$ | | 539-82-2 | 130.18 | 4.7450 |
| 14 | Citronellal $C_{10}H_{18}O$ | | 106-23-0 | 154.25 | 0.2800 |
| 15 | Isobutyl propionate $C_7H_{14}O_2$ | | 540-42-1 | 130.18 | 6.4700 |
| 16 | Allyl butyrate $C_7H_{12}O_2$ | | 2051-78-7 | 128.17 | 4.4430 |
| 17 | Methyl propionate $C_4H_8O_2$ | | 554-12-1 | 88.11 | 142.7780 |
| 18 | Pentyl acetate $C_7H_{14}O_2$ | | 628-63-7 | 130.19 | 3.5000 |
| 19 | Valeric acid $C_5H_{10}O_2$ | | 109-52-4 | 102.13 | 0.4520 |
| 20 | (+)Carvone $C_{10}H_{14}O_2$ | | 2244-16-8 | 150.22 | 0.0660 |
| 21 | (-)Carvone $C_{10}H_{14}O_2$ | | 6485-40-1 | 150.22 | 0.4000 |
| 22 | 2-Methoxypyrazine $C_5H_6N_2O$ | | 3149-28-8 | 110.12 | 4.2370 |
| 23 | Isoeugenol $C_{10}H_{12}O_2$ | | 97-54-1 | 164.20 | 0.0100 |
| 24 | Butyl acetate $C_6H_{12}O_2$ | | 123-86-4 | 116.16 | 11.5000 |
| 25 | Valeraldehyde $C_5H_{10}O$ | | 110-62-3 | 86.13 | 31.7920 |
| 26 | Isoamyl acetate $C_7H_{14}O_2$ | | 123-92-2 | 130.19 | 5.6000 |
| 27 | Methyl valerate $C_6H_{12}O_2$ | | 624-24-8 | 116.16 | 11.0430 |
| 28 | Octanal $C_8H_{16}O$ | | 124-13-0 | 128.21 | 2.0680 |
| 29 | 2-Hexanone $C_6H_{12}O$ | | 591-78-6 | 100.16 | 11.6000 |
| 30 | Methyl butyrate $C_5H_{10}O_2$ | | 623-42-7 | 102.13 | 31.1360 |
| 31 | 2-Heptanone $C_7H_{14}O$ | | 110-43-0 | 114.18 | 4.7320 |
| 32 | Acetophenone $C_8H_8O$ | | 98-86-2 | 120.15 | 0.3970 |

propionate, OSN mean $= 2.65 \pm 0.16$, $n = 42$, glomerular mean $= 2.53 \pm 0.32$, $n = 22$; MT + IP mixture, OSN mean $= 2.67 \pm 0.37$, $n = 48$, glomerular mean $= 2.26 \pm 0.22$, $n = 32$; $P > 0.05$ for all comparisons, Rank-sum test).

**Antagonism in complex odor blends**. After establishing that antagonism is a prevalent feature of odor mixture encoding in OSNs, we next wanted to understand the relationship between the complexity of an odor blend (that is, the number of elements in

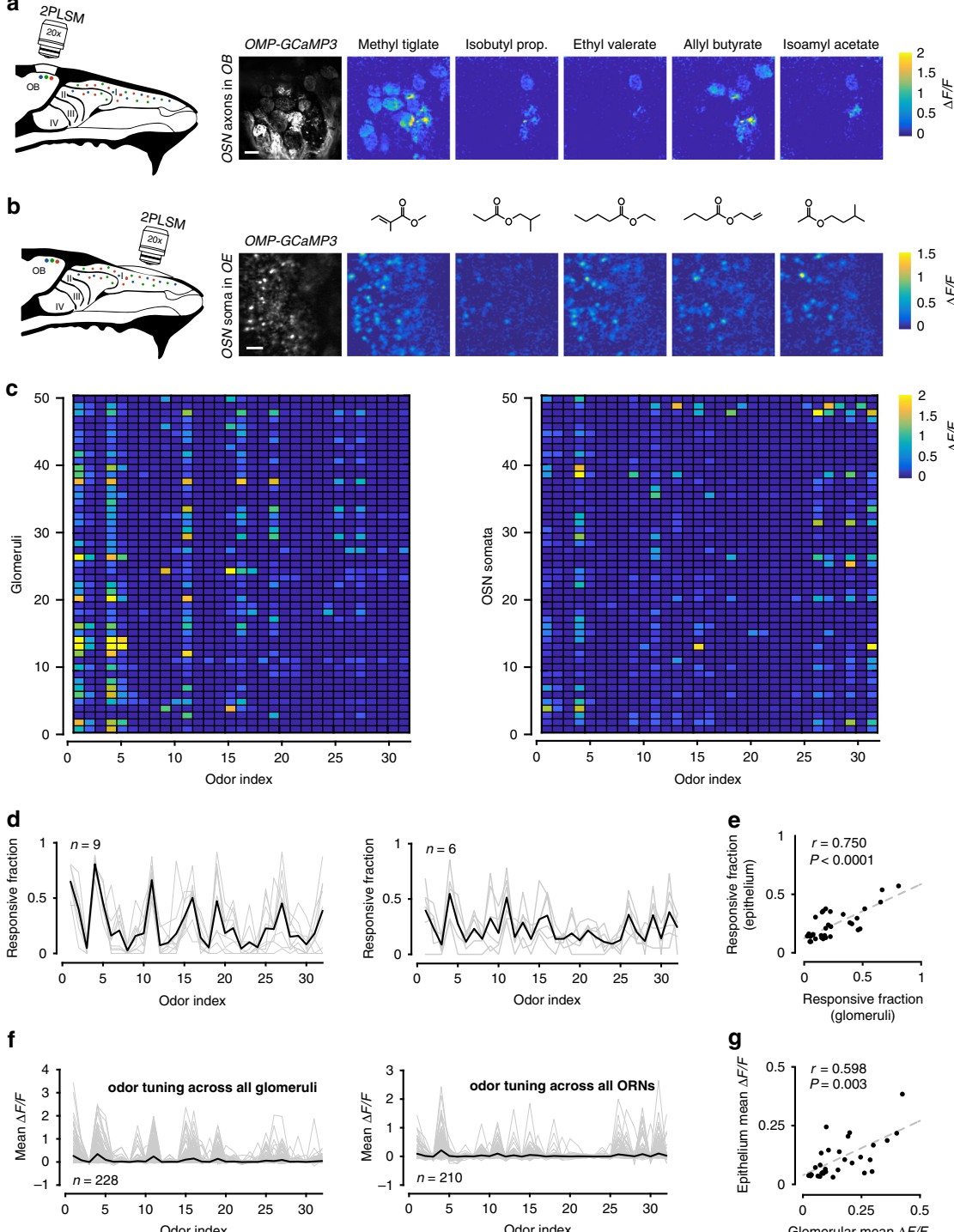

**Fig. 5 Odor tuning in the dorsal recess of the olfactory epithelium is similar to the dorsal olfactory bulb. a** Left, experimental setup for imaging axon terminals of OSNs in the glomerular layer of the olfactory bulb from one of five mice used. Right, resting fluorescence of glomeruli and heatmaps of selected odor responses. Scale bar is 100 μm. **b** Left, experimental setup for imaging OSN somata in the olfactory epithelium from one of six mice. Right, resting fluorescence in the soma of *GCaMP3*-expressing OSNs in the olfactory epithelium and heatmaps of selected odor responses. Scale bar is 30 μm. **c** Odor tuning of 50 randomly selected glomeruli (left) and 50 OSN (right) across 32 monomolecular odors. Odor index is consistent for each panel and follows Table 3. **d** Fraction of glomeruli (left; $n = 9$ imaging fields) or somata (right; $n = 6$ imaging fields) that responded to each of the odors in a given imaging field. Grey lines are individual fields and dark line is the mean across all imaging fields. **e** Scatter plot of the responsive fraction across all glomeruli and somata for each of the 32 odors used. Dashed line is the linear regression to the data (Pearson's correlation coefficient $r = 0.750$, $P < 0.0001$). **f** Odor tuning of 228 glomeruli (left) and 210 OSN somata (right). Grey lines are individual glomeruli or somata and dark line is the mean across all ROIs. **g** Scatter plot of the mean response across all glomeruli and somata for each of the 32 odors used. Dashed line is the linear regression to the data (Pearson's correlation coefficient $r = 0.598$, $P = 0.0003$).

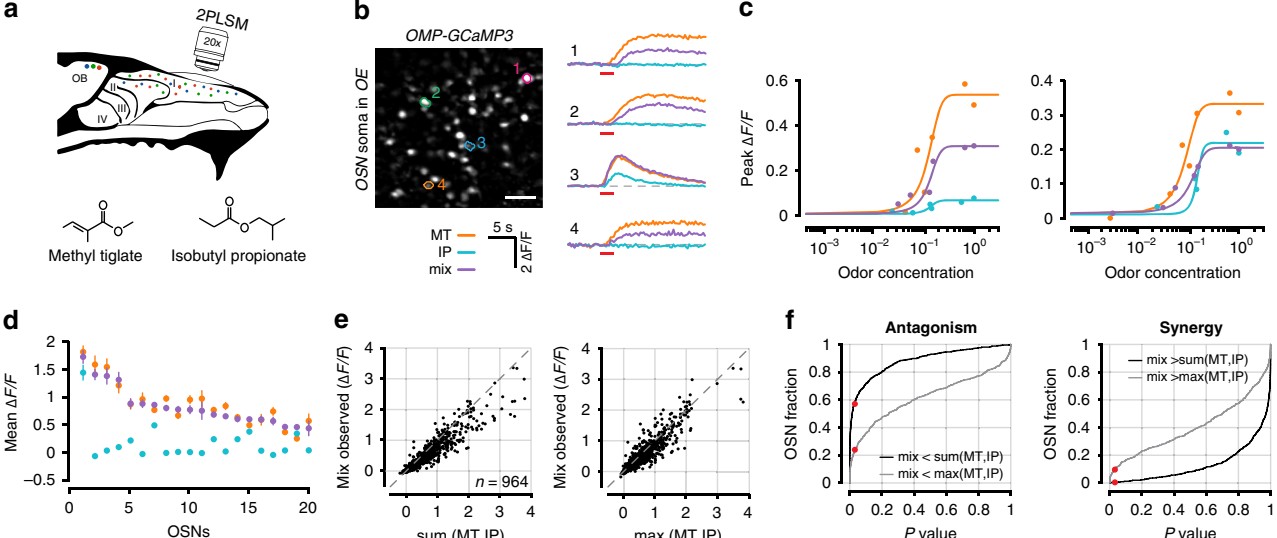

**Fig. 6 Antagonism in individual sensory neurons. a** Experimental setup for imaging OSN somata in the epithelium. **b** Example image of OSN somata and selected ROIs from one of six mice. Odor responses for two odors (Methyl tiglate and Isobutyl propionate) and a mixture of both odors from selected OSNs are shown on right as ΔF/F time courses. Red bar under traces denotes odor delivery time. Scale bar in image is 30 μm. **c** Example dose response curves from two selected OSNs. Each point is the average of 3–5 trials. **d** Data from 20 randomly selected OSNs. Each point is the mean response of the three highest odor concentrations for each odor. Error bars are SEM. **e** Left, scatter plot of the observed mixture response against the linear sum of both mixture components or (right) against the maximum response generated by either mixture component. **f** Left, cumulative distribution of one-sided rank-sum P values obtained for 964 OSNs for linear sum comparison (black) and comparison to maximum component response (grey). Red dots mark P values < 0.05. Right, cumulative distribution of left-sided rank-sum P values for linear sum comparison (black) and comparison to maximum component response (grey) to identify synergistic mixture interactions.

the mixture) and non-linearities in OSN responses. We imaged single OSN responses to 16 monomolecular odors (odor index 1–16 in Table 3), as well as random blends of 2, 4, 8, or 12 odors from this panel (Supplementary Table 1; Fig. 7c). When constructing complex odor mixtures, the concentration of each odor component was the same as when delivered alone. From the OSN responses to each of the single odors, we then made linear predictions for each of the mixture complexities. Our choice of odors for this experiment was motivated by several previous studies that demonstrate their efficacy and diversity in activating OSNs that presumably signal through GPCR signaling pathways[39–41], as well as past behavioral work[14,42], which demonstrates that mixtures of these odors can be perceived and distinguished by mice despite their overlapping OSN activity patterns.

For all mixture complexities, we routinely observed OSN responses that were far less than the linear prediction made by the summation of the individual mixture components (Fig. 7d). However, OSNs may never achieve the response predicted by linear summation, due to either firing rate or indicator saturation. To account for this possibility, for each OSN, we fit the data across all mixture complexities with a sigmoid with an initial slope of 1 and reaching an asymptote at the top 0.05 quantile of all observed mixture responses ($n = 100$). For each cell, we then calculated the deviation of each mixture response from the sigmoidal fit estimating the maximum response, or the linear prediction. We considered mixture responses that fell below the sigmoidal fit to be representative of antagonism (black line; Fig. 7d), while response that fell above the sigmoid represented synergistic interactions.

We observed nonlinear mixture responses in OSNs at all mixture complexities, although both the frequency and magnitude of such interactions increased with mixture complexity (Fig. 7f, $n = 1800$, $n = 1596$, $n = 1861$, $n = 1948$ OSN-odor mixture pairs, for 2-, 4-, 8-, and 12-part mixtures, respectively). For this analysis, we only considered OSN-mixture pairs where

the predicted OSN response was >0 ΔF/F. Importantly, mixture suppression was not the result of large trial-to-trial variability where a single trial strongly influenced the mean as the error for each mixture complexity was well below the sigmoidal fit to the data (Supplementary Fig. 3). For this analysis, we pooled the OSN responses for each mixture complexity and subsampled the distributions to test the significance of the observed nonlinearities against a null distribution. 200 random responses for each mixture complexity were selected and compared with distributions with a vanishing mean and the same standard deviation using a Kolmogorov–Smirnov test. This process was then repeated 100,000 times. Distributions of the P values obtained from these comparisons at each mixture complexity are shown on the right in Fig. 7f. For each mixture complexity we found >99% of all bootstrapped comparisons resulted in P values < 0.05.

To even more rigorously test our data, we performed an additional analysis where we compared OSN responses to odor mixtures against their maximum response to any of the mixture components alone (Supplementary Fig. 4). In close agreement with our prior analysis, we again found that likelihood, and the magnitude of mixture suppression in OSNs was related to the number of odor components contained within a mixture.

To exclude the possibility that a small number of cells most strongly contribute to the antagonism we observe, we normalized all measurements from each cell to the asymptote of the sigmoidal fit to the data and repeated our analysis. Here, we again observe that antagonism increases with mixture complexity (Fig. 7g). As a further confirmation of our finding, we repeated this experiment in OSN axon terminals in the glomerular layer. We collected data from 39 glomeruli in four mice. We again observed strongly nonlinear mixture interactions that increased with mixture complexity (Supplementary Fig. 5; $n = 673$, $n = 637$, $n = 733$, $n = 750$ glomeruli-odor mixture pairs, for 2, 4, 8, and 12-part mixtures respectively). Together, our results demonstrate that

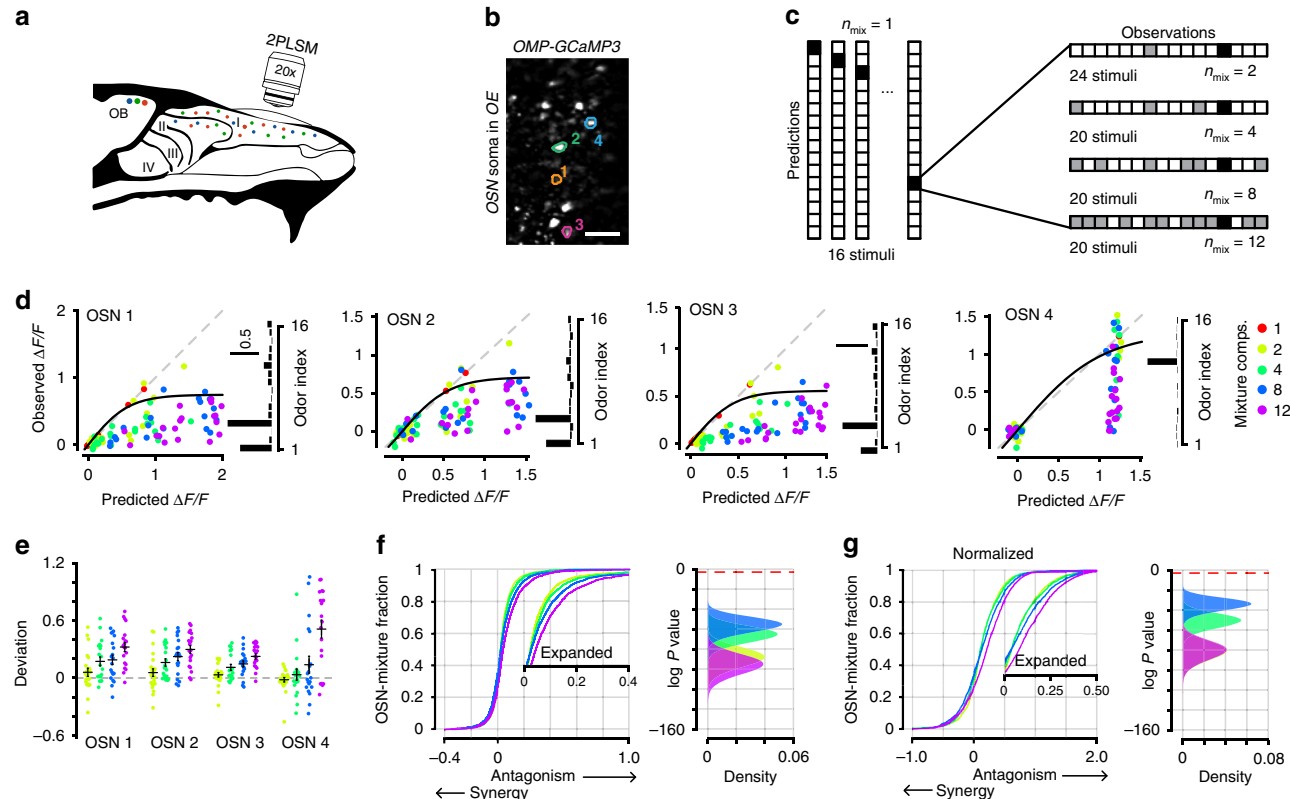

**Fig. 7 Antagonism through complex mixtures in single neurons. a** Experimental setup for imaging OSN somata in the epithelium. **b** Example image of OSN somata and selected ROIs from one of three mice used in this dataset. Scale bar in image is 20 μm. **c** Stimulus design for complex mixtures. OSN responses to each of 16 components are measured. Odor mixtures are made from subsets of 2, 4, 8, or 12 of the 16 individual odor components. **d** Example mixture responses from the OSNs outlined in (**b**). Each point is the average of three trials. The data from each OSN is fit with a sigmoid that reflects the maximum response from that cell. Odor tuning profile for each OSN is shown on the right. Scale bar units are ΔF/F. **e** Deviation from each mixture response to the sigmoidal fit the data. Positive values reflect antagonism and negative values reflect synergy. For each OSN: n = 24 2-part mixtures, n = 20 4-part mixtures, n = 20 8-part mixtures, and n = 20 12-part mixtures. Mean of all mixtures of a given complexity is the horizontal black bar and error is SEM. **f** Cumulative distribution of all deviations from the sigmoidal fit for each mixture complexity. Data collected from 129 OSNs in three mice. Expanded traces are inset. Right, histogram of bootstrapped P value distributions for each mixture complexity, testing the significance of the observed deviations from the sigmoidal fit against a null distribution. 500 random deviations for each mixture complexity were selected and compared with distributions with a vanishing mean and the same standard deviation using a two-sample Kolmogorov–Smirnov test. This process was repeated 100,000 times. Red horizonal dashed line corresponds to P values = 0.05. **g** Data were normalized for each OSN by dividing each mixture response by the asymptote of the sigmoidal fit to the data. Antagonism is not overrepresented in a small number of strongly responding cells.

input to the olfactory system may be normalized through mixture suppression of OSN activity.

## Discussion

It has been known for some time that individual odorant receptors can respond to multiple odors[23,43], so a reasonable question is how these receptors will respond to natural stimuli, which are almost always mixtures of odors. Odor mixture interactions have been described behaviorally for many decades[44–46]. One form of interaction is where binary (and larger) mixtures tend to have reduced perceived intensity (suppression). Odors can also be masked or harder to identify in mixtures, which could be due to reduced sensation of specific components. The origin of this perceptual odor suppression in mixtures is uncertain. Some of it could arise at the periphery, including at the receptors themselves. We recently proposed a simple model based on the two-step activation of odorant receptors, and noted that even a modest decorrelation of binding affinity and activation efficacy across odors could lead to antagonistic interactions, which will lead to mixture suppression[11]. In other words, odors that bind the most tightly to odor receptors may not strongly activate the odor

receptor complex and the downstream signaling cascade that ultimately generates OSN output.

Here, we have used direct imaging of OSN somata, as well as more conventional glomerular imaging, to show that mixture suppression (by inference antagonistic interactions) is widespread. We adapted a method first reported by Iwata et al.[34], to directly image the somata of OSNs in vivo, in freely breathing mice. We obtained responses of OSNs to a diverse array of odorants and found that the statistics of population responses were similar to that obtained for glomeruli. Interestingly, inhibitory responses were more easily detected in glomeruli than in OSN cell bodies, which might reflect glomerular averaging of heterogeneous responses of OSNs[47]. This OSN imaging method is likely to be valuable for understanding the fundamental properties of sensory coding and modulation at the periphery, especially with chronic imaging.

Previous studies using binary mixture stimuli have offered conflicting evidence for nonlinear interactions[17,24,48,49]. Evidence for relatively linear summation of odor responses to mixtures has been offered by studies that have imaged glomerular responses at low spatial and temporal resolution[48–50], but these studies did not systematically vary the concentration of odorants. Extracellular

spike recordings from single rat OSNs have indicated that a simple competitive binding model can explain the responses in about half the cases[17]. In the rest of the cases, more complex interactions including antagonism (competitive or non-competitive) and masking were needed to explain the observations[17]. Insect olfactory receptors also exhibit such nonlinear interactions[51,52]. Recently, we offered a principled explanation for all these different effects using a model that decouples the binding affinity of odorants to receptors and their activation efficiency[11]. Then, depending on the statistics of these properties in the repertoire of odorants encountered by an animal, the population of odorant receptors could exhibit varying degrees of antagonism or synergy.

In this work, we used binary mixtures and concentration variation over three orders of magnitude, to find that suppressive mixture interactions were remarkably widespread. We first demonstrated mixture suppression with glomerular imaging. Since glomerular responses are averages of many hundreds of OSN axons, they will overlook any heterogeneity. Glomerular responses may also be influenced by lateral interactions within the olfactory bulb networks, particularly due to GABAb-mediated presynaptic inhibition that can alter OSN calcium signals[33,53]. However, there is strong evidence that GABAb-mediated reduction of OSN responses are largely intraglomerular[30], which will simply serve as an automatic gain control, hence insensitive to the identity of the odorant. As an additional check, we chose two distinct odor mixtures with differing extents of overlap in glomerular activation. We found remarkably similar mixture effects for both pairs of odorants, which indicates that lateral interactions among glomeruli are unlikely to affect our conclusions.

To circumvent circuit and feedback interactions, we imaged OSNs directly and found that binary mixture suppression was just as widespread as observed with glomerular imaging. Since these interactions were observed at the earliest stages of odor encoding, in OSNs, the most likely source of mixture interactions is odorant receptors themselves. Evidence of antagonistic interactions in OSNs has been reported previously[16–18,54,55], and we advance these prior observations by providing evidence for widespread occurrence of this phenomenon. Going beyond binary mixtures, we used odor blends of increasing complexity, up to 12 components, to ask how mixture responses compared with the expected linear summation of individual responses (below saturation). This allowed us to ask how OSNs respond to more complex mixtures. It has been previously hypothesized that such nonlinearities in peripheral coding may be a mechanism to increase precision and specificity of encoding complex odor stimuli[11,12,17,54]. Indeed, using more naturalistic odor delivery conditions, we found that the degree of mixture suppression was greater with increasing number of components, which is expected from statistical consideration of antagonistic interactions[11]. This feature can be rationalized as increasing normalization of population responses, which we have shown previously to allow greater information transfer about odor identity, when saturation threatens degradation of information[11].

An important point regarding the odors we used in this study is that many of the odors share common molecular features (e.g., ester groups). A reasonable assumption is that there should be some molecular overlap between odors for antagonism to occur, due to the necessity of interactions between odor molecules and the binding pocket of olfactory receptors. However, in our panel we did include several odors that did not contain an ester group. When these non-ester odors were delivered as binary mixtures, in combination with ester containing odors, mixture suppression was still observed (Supplementary Fig. 6). These additional data, along with our previous observation that narrowly tuned ORNs show strong mixture suppression (Fig. 7d), argue that antagonism

is indeed a fundamental feature of mixture encoding in ORNs and is not a unique feature restricted to those that are sensitive to a single functional group.

Beyond our theoretical observations, there are other more recent experimental studies that report related effects[18–20]. Our work is unique in the following regard: (1) we report responses from a large set of OSNs in intact, freely breathing mice, responding to vapor phase odorants inhaled physiologically, (2) we report fast, real-time responses with a variety of odorants and using both OSN and glomerular imaging and (3) we report responses to mixtures of high complexity (up to 12 odors), well beyond just binary mixtures.

Given this widespread existence of normalization, what advantage would a similar phenomenon at the level of sensory transduction itself confer to the system? First order stimulus processing within the olfactory bulb is critically related to the activity of OSN inputs[56–59]. Having normalization occur early in the sensory hierarchy helps avoid saturation early on and preserve more information about the stimulus to be conveyed downstream. Indeed, in our previous work, we demonstrated using information theoretic calculations that a target odorant embedded in a complex mixture can be more easily detected with antagonistic interactions that lead to sparser representation[11]. The improved performance with antagonism exists for a wide range of receptor tuning widths (i.e., the average number of activated glomeruli per single odorant). Importantly, normalization at the level of receptors leading to sparser, more informative representation, comes for free without additional circuit burden.

## Methods

**Experimental model and subject details.** Adult heterozygous *OMP-GCaMP3* mice[60] of both sexes were used in this study. All animals were produced from a breeding stock maintained within Harvard University's Biological Research Infrastructure. All animals were between 20 and 30 g before surgery and singly housed following any surgical procedure. Animals were 2–6 months old at the time of the experiments. All mice used in this study were housed in an inverted 12 h light cycle and fed ad libitum. Animals were housed at 22 ± 1 °C at 30–70% humidity. All the experiments were performed in accordance with the guidelines set by the National Institutes of Health and approved by the Institutional Animal Care and Use Committee at Harvard University.

**Olfactory bulb craniotomy.** A craniotomy was performed to provide optical access to both olfactory bulbs. Mice were first anesthetized with an intraperitoneal injection ketamine and xylazine (100 and 10 mg/kg, respectively) and the eyes were covered with petroleum jelly to keep them lubricated. Body temperature was maintained at 37 °C by a heating pad. The scalp was shaved and then opened with a scalpel blade. After thorough cleaning and drying, the exposed skull was gently scratched with a blade, and a titanium custom-made headplate was glued on the scratches with Loctite 404 Quick Set Adhesive. The cranial bones over the OBs were then removed using a 3 mm diameter biopsy punch (Integra Miltex). The surface of the brain was cleared of debris. The surface of the brain was kept moist with artificial cerebrospinal fluid containing in mM (125 NaCl, 5 KCl, 10 Glucose, 10 HEPES, 2 CaCl2 and 2 MgSO4 [pH 7.4]) and Gelfoam (Patterson Veterinary). Two 3 mm No. 1 glass coverslips (Warner) were glued together with optical adhesive (Norland Optical Adhesive 61) and adhered to the edges of the vacated cavity in the skull with Vetbond (3 M). C&B-Metabond dental cement (Parkell, Inc.) was used to cover the headplate and form a well around the cranial window[61,62]. After surgery, mice were treated with carprofen (6 mg/kg) every 24 h and buprenorphine (0.1 mg/kg) every 12 h for 5 days. Animals were allowed to recover for at least 3 days. Prior to each imaging session, animals were anesthetized with a mixture of ketamine and xylazine (90% of dose used for surgery) and body temperature was maintained at 37 °C by a heating pad. Respiration was measured through an airflow sensor (Honeywell)[63] during most experiments and maintained between 0.5 and 1.5 Hz (traces in Supplementary Fig. 1f).

**Olfactory epithelium thinned skull procedure.** Mice were anesthetized using the same procedure and all pre-surgical methods through head plate implantation are the same as the craniotomy. The cranial bones over the olfactory epithelium were thinned with a dental drill and blade until transparent[34]. The thinned area of skull was then covered with cyanoacrylate adhesive (Loctite) and a class coverslip was implanted in the adhesive. Dental cement was then used to form a well over the

thinned section of skull. Following a recovery period, the thinned bone procedure allowed for chronic imaging of the epithelium for several weeks.

**Multiphoton Imaging**. A custom-built two-photon microscope was used for in vivo imaging. Fluorophores were excited and imaged with a water immersion objective (20×, 0.95 NA, Olympus) at 920 nm using a Ti:Sapphire laser (Mai Tai HP, Spectra-Physics). Images were acquired at 16-bit resolution and 4–8 frames/s. The pixel size was 0.6 μm OSN somata imaging and 1.2–2.4 μm for imaging glomeruli. Fields of view ranged from 180 × 180 μm in the epithelium to 720 × 720 μm in the glomerular layer. The point-spread function of the microscope was measured to be 0.51 × 0.48 × 2.12 μm. Image acquisition and scanning were controlled by custom-written software in LabView (National Instruments). Emitted light was routed through two dichroic mirrors (680dcxr, Chroma and FF555- Di02, Semrock) and collected by a photomultiplier tube (R3896, Hamamatsu) using filters in the 500–550 nm range (FF01–525/50, Semrock).

**Odor stimulation**. Monomolecular odorants (Sigma or Penta Manufacturing) were used as stimuli and delivered by custom-built 16 channel olfactometers controlled by custom-written software in LabView[61]. For binary mixture experiments, the initial concentration series for each odor was between 0.08 and 80% (v/v) in mineral oil and further diluted 16 times with air. For all experiments, the airflow to the animal was held constant at 100 mL/min and odors were injected into a carrier stream. The relative odor concentration was measured by a photoionization detector (PID; Aurora Scientific) and normalized to the largest detected signal for each odor. To create mixtures, air-phase dilution was used, and the total concentration of each odor was held constant. The measured mixture signal in the PID was nearly a perfect linear summation of the signal measured for each odor alone (Supplementary Fig. 7). For all experiments, odors were delivered 2–6 times each.

For experiments characterizing the odor tuning of olfactory epithelium, the odor panel consisted of: (1) Ethyl tiglate (2) Allyl tiglate (3) Hexyl tiglate (4) Methyl tiglate (5) Isopropyl tiglate (6) Citronellyl tiglate (7) Benzyl tiglate (8) Phenylethyl tiglate (9) Ethyl propionate (10) 2-Ethyl hexanal (11) Propyl acetate (12) 4-Allyl anisole (13) Ethyl valerate (14) Citronellal (15) Isobutyl propionate (16) Allyl butyrate (17) Methyl propionate (18) Pentyl acetate (19) Valeric acid (20) (+)Carvone (21) (−)Carvone (22) 2-Methoxypyrazine (23) Isoeugenol (24) Butyl acetate (25) Valeraldehyde (26) Isoamyl acetate (27) Methyl valerate (28) Octanal (29) 2-Hexanone (30) Methyl butyrate (31) 2-Heptanone (32) Acetophenone. See Supplementary Fig. 7 for PID measurements. For experiments measuring complex mixture responses in the olfactory epithelium, odors 1–16 were used from the panel above. Additional odor information is available in Table 3 and Supplementary Table 1.

**Data analysis**. Images were processed using both custom and available MATLAB (Mathworks) scripts. Motion artifact compensation and denoising was done using NoRMCorre[64]. For experiments imaging OSN axon terminals in the olfactory bulb, regions of interest (ROIs) were manually selected by outlining glomeruli in maximum projection images. For epithelium imaging, the CaImAn CNMF pipeline[65] was used to select and demix ROIs. ROIs were further filtered by size and shape to remove merged cells. For all mixture experiments, the peak ΔF/F signal was calculated by finding the peak signal following odor onset and averaging with the two adjacent points. The mean ΔF/F signal in the 20 frames following odor onset was used for odor tuning experiments. To account for changes in respiration frequency and anesthesia depth, correlated variability across replicates was corrected for[14]. Thresholds for classifying responding ROIs were determined from a noise distribution of blank (no odor) trials from which three standard deviations were used for responses. Across all datasets, only ROIs with at least one significant odor response were included for further analysis. Measurements of binary mixture nonlinearities used individual trial replicates of the three highest odor concentrations used in each experiment.

**Data fitting**. We further analyzed the response curves from 344 glomeruli corresponding to Methyl tiglate, Isopropyl propionate, and their mixture, 226 glomeruli corresponding to Ethyl valerate, Allyl butyrate, and their mixture, and 964 ORNs corresponding to Methyl tiglate, Isopropyl propionate and their mixture. The observed $Ca^{2+}$ fluorescence response $R(c)$ was fit using a sigmoid function against log odor concentrations delivered over three orders of magnitude using the equation:

$$R(c) = b + R_{max}/(1 + e^{-n(\log c - \log K)}). \tag{1}$$

The four free parameters that are fit to the data are $b$, $R_{max}$, $K$ and $n$. Here $b$ is interpreted as the baseline response for blank odors, $R_{max}$ is the response above baseline at saturating concentrations, $K$ (commonly referred to as the dissociation constant) is the reciprocal of the binding affinity and $n$ is the Hill coefficient. To account for noise and variability in the data, we used four criteria to select the fitted curves (all units are ΔF/F): (1) The total squared deviation of the data from the best fit sigmoid shouldn't be too large. We used a threshold of 0.1 to ensure rejection of highly noisy data. (2) The maximum value over all concentrations should be >0.4 ΔF/F, (3) Binding affinity should be within the range of the concentrations used

and (4) the hill coefficient cannot be below 0.75 or above 6 to exclude unreasonably sharp fits.

**Reporting summary**. Further information on research design is available in the Nature Research Reporting Summary linked to this article.

## Data availability
The data that support the findings of this study are available from the corresponding author upon reasonable request.

## Code availability
The code used for analysis and figure generation of this study are available from the corresponding author upon reasonable request.

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

## Acknowledgements

This work was partly supported by grants from the NIH (R01 DC014453 & DC016289) to VNM. JDZ was supported by NIH Fellowships F32 DC015938 & K99 DC017754. GR was partially supported by the NSF-Simons Center for Mathematical & Statistical Analysis of Biology at Harvard (award number #1764269) and the Harvard Quantitative Biology Initiative. We also acknowledge support from the following grants, which facilitated critical discussions at KITP, Santa Barbara: NSF (PHY-1748958), NIH (R25 GM067110), and the Gordon and Betty Moore Foundation (2919.01). We would also like to thank members of the Murthy Lab for helpful discussions.

## Author contributions

J.D.Z., G.R., M.V., and V.N.M. designed the experiments. J.D.Z. acquired the data. J.D.Z. and G.R. analyzed the data. J.D.Z. and V.N.M. wrote the paper with inputs from all authors.

## Competing interests

The authors declare no competing interests.
