## [Peer Review File · Nature Communications]

Reviewers' Comments:

Reviewer #1:

Remarks to the Author:

Zak et al present evidence supporting their previous theoretical work suggesting antagonistic interactions at the level of odor transduction. Dissociating binding affinity and activation efficacy can give rise to these effects. These effects must be highly relevant to olfactory perception, at the least because they prevent saturation in responses to complex odor blends, as the authors propose. I suspect that these interactions are also highly relevant to Rokni's work on detecting components within figures, as well as the well-established configural nature of olfactory perception. The major conclusions of the paper are:

1) Pairwise antagonistic interactions are common, even at the level of OSN somata, where synaptic interactions cannot contribute to the observed effects (in contrast to the more typical configuration of imaging OSN axon terminals, which do receive synaptic input). 20-60% of OSN somata show this antagonism, depending on the criterion. This phenomenon must have a major impact on odor-driven behavior.

2) Antagonism becomes more pronounced in blends with more odor components. I'm not sure about how the data supporting this conclusion are quantified. If I understand correctly, the authors fit a sigmoid to the largest odor responses, and then quantified the deviation of real data from that sigmoidal function. I get that this measure was chosen in part to allay the concern that sublinear effects may reflect firing rate or indicator saturation, but this still seems to be an odd choice. Perhaps something simpler could make the point more clearly? Given that the example OSNs in 5D only appear to respond to 1-2 odors, it might be useful to plot mixture responses normalized to the max response (similar idea to plotting observed against the max response in the earlier figures). If reduction from this max is proportional to the number of components in the mixture, the conclusion is supported.

Minor point: what is the take-home message of 3Ei and ii? It is sub-optimal to plot response magnitude with circle diameter when the vertical separation of points on the y-axis is less than the diameter -- big circles hide little circles. If the point is to represent the distribution of response magnitudes across the population of odor-OSN/glom pairs, a colormap could preserve more of the information. I.e., a 50 X 32 grid where the colormap represents the response magnitude.

This work is rigorous and relevant, and will influence the field to think more about events at the sensory surface, and not always assume that interesting response properties reflect circuit events.

Sincerely,
Matt Smear

Reviewer #2:

Remarks to the Author:

There have been several previous reports of antagonistic odor interactions in the peripheral olfactory system. However, these findings have typically been limited to specific odor-receptor pairings and considered to be evidence of special exceptions to the general rule that odors interact largely additively in the periphery. However, the present study presents strong neurophysiological evidence that antagonistic interactions in the periphery are actually the norm in the olfactory system. This has profound implications for essentially all models of olfactory coding in the brain. The authors do not attempt to explicate the mechanism of this interaction, which would be a big project requiring a

different toolbox. However, in this reviewer's opinion the rich demonstration of the phenomenon is sufficient to drive a high impact for this paper as it is. The work itself is performed with an appropriate degree of scientific rigor and the methods are established enough to permit ready reproducibility. My suggestions for improvement are relatively minor.

General comments

The potential complexity of odor coding in the periphery beyond single parameter affinity models is an underconsidered question that is suddenly attracting substantial interest, and the authors make an important contribution to our understanding of what actually happens in vivo and its utility for interpreting odor mixtures.

The core finding that odor pairings can produce antagonistic interactions at the level of the OSNs is nicely demonstrated, including important analyses that demonstrate that the finding holds across odor concentrations.

The diversity of Hill function parameters across glomeruli is an interesting element of the data that is underexplored. It would be helpful to see a histogram or scatter (perhaps with error bars showing individual confidence) of estimated Hill function coefficients in Figures 1 and 2. They would not be expected to be the same in every glomerulus and would be inform consideration of the population-level dynamic range of each OSN subtype, which seems relevant to mixture coding in this context.

The authors effectively combine in vivo population-level data from OSN axons at their convergence in the olfactory bulb with imaging of individual OSN cell bodies in the epithelium and observe qualitatively similar effects. This difficult step is essential to ruling out inhibitory circuit effects in the presynaptic terminal.

The heterogeneous but reliable kinetics of single cell odor responses is an interesting issue that is relegated to a supplementary figure and a single sentence in the results. Though such data are always somewhat challenging to interpret because of the kinetics of GCaMP itself, the heterogeneity deserves richer consideration in the main text. In particular, one key analysis is missing: does the degree of antagonism vary across the duration of the odor-evoked response, or is it a constant proportional reduction over time? How does this heterogeneity alter the population response across populations of glomeruli or cells? Any temporal differences would have important implications for mixture coding and also might shed light on potential mechanisms of the antagonism.

It remains possible that the degree of antagonism observed is somehow especially common for ester odorants, which are utilized for practical reasons (they tend to activate neurons the right place to permit imaging), and thus the otherwise arbitrary choice of odorants might lead to unrepresentative data. However, this seems unlikely, both because the authors do include some evidence that the key effect can happen with non-ester odorants in supplementary data and because there are many previous studies that used esters as part of a larger panel of odors with no evidence that they behave any differently than other chemical classes.

Odor mixtures were generated in air phase and their effective concentrations were validated using a photoionization detector. This is an important element of technical rigor for these experiments.

There is a previous report that OSNs in male and female mice may exhibit different degrees of odorant selectivity (Kass 2017 in Scientific Reports). Because the present study used both male and female mice, a post-hoc analysis of the present data to test for evidence of a sex difference could be informative.

Reviewer #3:

Remarks to the Author:

The paper of Zak and colleagues deals with the fundamental mechanisms engaged since the first two steps of the olfactory system in encoding of olfactory stimuli, which are mainly complex odorant mixtures. The work of Zak and colleagues present several fundamental qualities: first, their technical expertise in calcium activity imaging sounds impressive and allow them to work in vivo in freely breathing mammals, mice, second they make their recordings both at the olfactory sensory neuron (OSN) level of the dorsal recess of the sensory mucosa (967 units!) and the glomerular level (334+226 glomeruli!) and third they used complex stimuli, i.e. odorant mixture (up to 12 compounds) in attempting to describe the first two stages of the system, by working in conditions as close as possible as physiological ones. Such comparative recordings allow the authors to constate that all the antagonistic odorant-odorant interactions in mixture they observe at the glomerular level are already observable from the individual ORNs' responses; so the authors propose they are mainly inherent in the integrative qualities of OSNs, thus do not result from presynaptic or intra-, inter-glomerular synaptic activities. Noticeably is that glomerular responses which quantitatively amplify and integrate the activity of OSNs endowed with same olfactory molecular receptor (OR) mainly show suppressive interactions between odorants even if one of them, in the case of binary mixture, did not elicit a clear excitation (such results have been described in Chaput et al., 2012 in rats in vivo in European journal of neurosciences and in Ukhanov et al, 2010; 2011, in vitro in Journal of neurophysiology). This reveals silent suppressive interaction which can be also observed from unit OSN responses. More generally, the dominating dominant character of antagonistic (= suppressive) interactions between odorant in mixture is a crucial result which enlighten the main mechanisms from which, for a given odorant mixture, is generated a unique and precise OSN response pattern; the more complex the mixture the more precise the pattern will be; and this despite the low selectivity of ORs and the infinity of stimuli constituting the odor world. The method section sounds well, supplementary figures are useful. The experiments seem to be well done, well though. Such a paper makes it possible to decisively advance concepts that were hitherto fragmented.

To my opinion the work of Zak and co-workers is highly important and fundamental for the advancement of knowledge in the understanding of the olfactory peripheral integration of odorant stimuli. The one slight shadow I see in this study is the choice of stimuli. However, this is largely due to the fact that this choice is not motivated, or introduced as being made on based or preconceived hypotheses. I suggest to the authors to take some lines to motivate their choice regarding some of their preceding studies or bibliography in general. A very important and positive point that the authors used gaseous stimuli and make mixtures in gaseous phase, it is really fundamental because this obliges de facto to take into account the differences in volatility between the odorants, what the system is designed to do when it is facing the real odorant world. it is enough fundamental to further describe them I also think it is fundamental to give a table with stimuli, CAS number, molecular Weight, SVP and measures regarding theoretical calculation from liquid phase. May I leave the authors know that to my opinion; the next decisive step would be to take real mixtures with chromatographic decomposition and to record neural responses in mirror of the measured concentrations and ratios of compounds. This likely allow to describe the successive and progressive steps of the OSN pattern construction and to enlighten the role of the "silent" compound(s) (if there are some) on the final pattern.

As a conclusion, I highly recommend the publication of this work

Remarks and questions throughout the text:

Summary: I sincerely think that the summary might be improved, it does not do justice to the work. the results could be better showcased. I don't like the term "normalization" which makes me think that the extremely complex integration mechanisms set in motion at different levels, docking, binding, signalling pathways... simply result in a normalization, which implies that saturation is thus avoided. Maybe, first "normalization " is a false friend regarding the French word "normalisation", maybe second, to sum up peripheral treatment to a mechanism simply put in place to avoid saturation seems to me to be restrictive and not very rewarding. Anyway I advise the authors to change the summary in this direction.

L142: odor tuning should be changed in odor tuning.

L226: Please add ref 23 or our paper in Science, it was the first work in the rat and comparatively to Malnic et al, it was an in vivo study. For the small history, before this study in rats, the fact that each individual OSN was responsive to many monomolecular odorants had been firstly demonstrated in the seventies, in frogs in the lab I made my thesis (which was in frog too). During a congress in Sienna in september 1998, I remember that Richard Axel said to me that frogs were not rats and that, about my results in rats, electrophysiologists do not not correctly work since they use stimulus at huge concentrations which explain their results where unit OSN displays a wide odorant profile.

L234: May the authors further explain what do they mean by decorrelation because even if the authors published that result before this could help here to better understand.

L244: What do the authors mean by "plasticity" here, the presence of this word needs to be developed a little more or remove.

L244: through "chronic" the authors mean in vivo or during several sessions?

L270-282: Even if they go beyond binary mixtures, may I suggest to the authors that this discussion about OSNs can comprise a little bit more citations of some literature dealing with antagonistic interactions with binary mixtures at OSN level. Especially the authors should refer to Chaput et al., 2012.

As a matter of fact, the authors write "Antagonistic interactions have been hinted at previously, but we find direct evidence for widespread occurrence of this phenomenon."

I don't fully agree with the verb "hint" Antagonistic interaction have been demonstrated though unit OSN recordings in vivo in rats, even if the in vivo results prior to their study remain poor in number.

L278: Maybe the authors could discuss this fundamental result which was until now only an hypothesis in term of peripheral coding which increases in precision and specificity with odor stimulus complexity, finally here the system works in really natural condition by comparison with experiments which used monomolecular or binary mixtures. So, in physiological condition, the system gives proofs of a great specificity since the periphery.

In a more general way, I suggest the authors to discuss about the choices they have made to build mixture and the importance for a future study to test realistic mixtures, not only experimental ones. Indeed, in the real chemical world, compounds of a mixture highly compete to volatilize because they are rarely at equalized ratios, have not the same saturated vapor pressure; to my opinion the olfactory system works in its plenary potentialities only confronted with such realistic mixtures. With the experimental ones, the system does not work under optimal conditions although not conditions not as bad as with monomolecular odorants. Despite the authors use "experimental" and not realistic mixtures they have respected a crucial point by taking care to use the same concentrations when they compare responses to individual compounds and to their mixtures, and they have checked that mixture concentration was the linear sum of each compound concentration.

L289-292; "These additional data, along with our previous observation that narrowly tuned ORNs show strong mixture-suppression (Figure 291 5D), argue that antagonism is indeed a fundamental feature of mixture encoding in ORNs and is 292 not a unique feature restricted to those that are sensitive to a single functional group."

Here, to my opinion, it is a crucial result that highly indicates that the very precise organization of the

projection of OSNs onto glomeruli based on the OR expression ensures the maintenance of the hardware organization but cannot be directly and simply linked to a functional spatial coding; what needed to be demonstrated.

L283-289: This result that stimuli with close chemical features involve suppressive interactions was very important and could be discussed regarding the seminal results obtained in frog (group of Duchamp, Holley, Revial and Sicard in the seventies) with monomolecular stimuli which established qualitative space by comparing the unit qualitative profiles by pairs. What do you think about these results? Of course, they have been obtained in "physiological extreme conditions" since the system is not design to process monomolecular odorant but put in light of results obtained with complex blends, it could bring a new look on the qualitative groups established from monomolecular stimuli and further strengthen the authors' contribution.

L293-298: I fully agree with you. You should add: we report responses from a very large set of OSNs!

L300: I cannot see the widespread suppression as a normalization (but may I don't well catch the meaning of the word), this word is negative for me. Of course, I agree with the authors the system must be preserve from saturation but it is more than that. To my opinion the olfactory system makes exactly the contrary of the normalization, more complex is a stimulus, more compounds compose it, more precise et unique will be the OSN combination encoding it (moreover a plausible hypothesis is that fewer neurons are involved when the complexity increases) to my idea this does not match with the notion of normalization.

L328: To my view, it should be added that the surgical procedure allows to record the calcium activity of OSNs lying in the dorsal recess region of the OE. From the procedure description of the OB craniotomy I understand that the same mice can be recorded several times. Is it correct? On the other hand, for the OSN recordings, I guess that the authors make only one recording session? Is it correct? It should be better to clarify and to precise, if it is the case, that surgical procedure of the nose does not allow to keep the animal implanted to be chronically recorded.

L346: Odor stimulation:

The best would be that the authors give a table gathering the odorants, such a table including CAS number of molecules, the saturating vapor pressure, molecular weight and concentrations used, the same applies to the mixtures, for wich are needed the list of components, ratio and fraction of each and, in addition given that each stimulus has not the same volatility properties, the range over which each molecule was used. I think such information is important and fundamental to allow for future comparisons.

L351: "The absolute relative odor concentration was measured by a photoionization detector (PID; Aurora Scientific) and normalized to the largest detected signal for each odor "

It is fine to make such measures and to make mixtures in air-phase.

Please what does mean the absolute relative concentration? (the same question holds for the legend of Fig.1)

Please what does mean "normalized to the largest signal for each odor?"

L355: It is very fine that the authors could check that quantitatively the mixture is the linear summation of each odorant alone. I would be perfect if the authors precise the chosen ratio of each compound; it ca be don in the table suggested before.

L357: Given the fundamental importance of the choice of odorant stimuli, can the authors say a little bit more about the choice of them regarding literature? Preceding experiments? Their effectiveness at the periphery? Etc? Why did the authors select odorants from 1 to 16 for complex mixture stimuli?

L389: Can the authors describe more precisely the equation parameters of the response? the authors define RC, b and R_{max+k} and $k-1$ which does not appear in the equation. May the authors clarify please.?

Fig5: Fig5C: the authors should clarify what do they mean by predictions, why only one box is blackened ? Is it the best stimulus? Do the grey boxes mean that the responses to thes stimuli were weaker? White boxes are for stimuli that did not elicit responses What are the individual odorants?

Give the composition of mixture, components, ratio, PID measures if the authors had? Maybe all this information can be in the stimulus table.

Supplemental Fig2: This figure is really very interesting. Please can the authors precise the constitution of mixtures? Are there the same for the 5 OSNs?

Supplemental Fig 5: A: PID measures are very useful, for example in A to compare the compound volatilities, we can observe here that Methyl tiglate and isobutyl prop. Are not very far from each other and that their mixture is really the summation of them.

C: What do the values to the left of the PID traces for the odorants correspond to i.e. to which concentration in % vol/vol, is it the same for each odorant?

Lyon on 1st April 2020

Patricia Duchamp-Viret

Reviewer response to NCOMMS-20-06579, “Antagonistic odor interactions in olfactory sensory neurons are widespread in freely breathing mice.”

We thank each of the reviewers for their insightful comments and constructive suggestions. Below consider each comment and explain our revision or provide further explanations as requested. In the manuscript file, major changes are highlighted in red text. For convenience, where applicable we have pulled text from the manuscript and placed it directly in this document. Below, reviewer comments are in bold typeface.

Reviewer 1

We would like to thank Dr. Smear for his positive comments and suggestions.

1) “It might be useful to plot mixture responses normalized to the max response (similar idea to plotting observed against the max response in the earlier figures). If reduction from this max is proportional to the number of components in the mixture, the conclusion is supported”.

This is an excellent suggestion; however, we feel that our approach aligns with the spirit of Dr. Smear’s suggestion, and is also a more conservative normalization method. We would like to draw attention to Figure 5g, where we have performed the analysis that the reviewer suggests above. Perhaps the accompanying text does not fully and clearly illustrate our analytical approach. We have revised the text to more fully describe our analysis, but for the consideration of the reviewer, below is a summarized version.

For each OSN, the sigmoid asymptote was fit to the top 0.05 quantile of the mixture responses – this is a more conservative method than normalizing to the absolute max and helps account for the intrinsically noisy imaging data. The CDF plots in Figure 5g show that when the responses from each OSN are normalized this way, the reduction from the sigmoidal bound is indeed proportional to the number of components in the mixture – the larger mixture CDF plots fall further to the right.

It is possible that we have misinterpreted the suggestion and what the reviewer actually means is that the mixture response could be compared to the maximum response of any of its constituent components. We have now performed this analysis as suggested (see new Supplementary Figure 4). One other point of clarification is that we cannot directly normalize to the max response because maximum responses that are near 0 df/f will dramatically skew the results. Therefore, for this analysis, we compare the distance of each mixture from the maximum response of any of the mixture components.

2) “what is the take-home message of 3Ei and ii?... If the point is to represent the distribution of response magnitudes across the population of odor-OSN/glom pairs, a colormap could preserve more of the information”.

We agree with this reviewer point and we have changed the plots in *new* Figure 5e to be represented by a color scale rather than a dot plot.

Reviewer 2

We also thank Reviewer 2 for her/his positive comments and recommendations.

1) The diversity of Hill function parameters across glomeruli is an interesting element of the data that is underexplored. It would be helpful to see a histogram or scatter (perhaps with error bars showing individual confidence) of estimated Hill function coefficients in Figures 1 and 2.

We agree that this is an interesting point to convey to readers and have now added new information below to the methods.

“To account for noise and variability in the data, we used four criteria to select the fitted curves (all units are df/f): 1) The total squared deviation of the data from the best fit sigmoid shouldn't be too large. We used a threshold of 0.1 to ensure rejection of highly noisy data. 2) The maximum value over all concentrations should be greater than 0.4 df/f, 3) Binding affinity should be within the range of the concentrations used and 4) the hill coefficient cannot be below 0.75 or above 6 to exclude unreasonably sharp fits.”

We now include a new Figure 4 that displays each fit that satisfied the above criteria, as well as the distribution of Hill coefficients. Bottomline: we find that the parameters are not significantly different as a function of odor or neuron/glomerular identity.

2) The heterogeneous but reliable kinetics of single cell odor responses is an interesting issue that is relegated to a supplementary figure and a single sentence in the results.

We agree with the reviewer that the OSN kinetic heterogeneity is indeed an interesting finding. However, given the complexity of calcium dynamics in OSNs, we cannot make strong statements on the source of the response kinetic heterogeneity. For instance, such effects could arise from, indicator expression, receptor expression levels, differential cAMP production, other effects related to downstream Ca-activated chloride channels, and calcium buffering/extrusion properties, to name a few possibilities. As we see it, the main point of this figure is to show that the kinetics are reliable from trial to trial – an important consideration for our mixture experiments.

3) Does the degree of antagonism vary across the duration of the odor-evoked response, or is it a constant proportional reduction over time?

For the reasons described above, it is extremely difficult to make any mechanistic statement about calcium signals and their relationship to the OSN output beyond the peak signal, which is likely to be proportional to the maximum firing rate of a given neuron. Furthermore, our recently published model (Reddy, Zak et al., 2018, *eLife*) which provides the theoretical motivation for the experiments performed in this manuscript, predicts that the peak firing rate should be strongly modulated by antagonistic interactions. Although, we would like to point out that we do agree with the reviewer's sentiment that the entire spike train generated by an OSN should be influenced by antagonism; however, the effects will be most pronounced at the peak of the $\Delta F/F$ signal, thus motivating our focus on this feature of the signals.

4) Because the present study used both male and female mice, a post-hoc analysis of the present data to test for evidence of a sex difference could be informative.

While we used mice of both sexes for this study, we did not analyze our data with respect to sex and in many cases our datasets are not large enough for a proper analysis. For instance, the dataset of complex mixtures (Figure 7) is taken from 3 animals, therefore a proper sex comparison cannot be made. We agree with the reviewer that this would indeed be an interesting and potentially impactful analysis; however, our studies were not designed with this type of analysis in mind and therefore such a question is beyond the scope of the present manuscript.

Reviewer 3

We thank Dr. Viret for her thoughtful and thorough comments on our manuscript.

Summary: I sincerely think that the summary might be improved, it does not do justice to the work. the results could be better showcased. I don't like the term "normalization" which makes me think that the extremely complex integration mechanisms set in motion at different levels, docking, binding, signaling pathways... simply result in a normalization, which implies that saturation is thus avoided. Maybe, first "normalization" is a false friend regarding the French word "normalisation", maybe second, to sum up peripheral treatment to a mechanism simply put in place to avoid saturation seems to me to be restrictive and not very rewarding. Anyway, I advertise the authors to change the summary in this direction.

We understand and appreciate the intent of Dr. Viret's comment; however, we would like to emphasize "normalization" is a technical term related to processing of sensory inputs. As defined by Carandini & Heeger in their 2012 review on the topic (Nat Rev Neuroci 13:51-62), "... normalization, in which the responses of neurons are divided by a common factor that typically includes the summed activity of a pool of neurons". "The computational benefits of normalization include maximizing sensitivity, as well as facilitating the discrimination among representations of different stimuli." Our (exciting) point here, and in our theoretical predecessor paper (Reddy, Zak et al eLife 2018), is that this important computational feature already exists at the level of OSNs, even before circuitry is involved!

We wish to point out that we do not imply that normalization relegates the activity of the OSN complement to an indiscriminate uniform level (which may be the cause of Dr. Viret's concern!), but rather functions to prevent saturation of the input layer in the olfactory system. But of course, this may not be the ONLY function of antagonistic interactions, but one which we have chosen to focus on, based in part, on our prior work. Furthermore, we have expanded our abstract and elsewhere in the manuscript to emphasize that our study is not just singularly focused on normalization, but also information transfer in general.

L142: odor turning should changed in odor tuning.

This is now corrected. Thank you for catching our mistake.

L226: Please add ref 23 or our paper in Science.

This important reference has been added, we apologize for the oversight.

L234: May the authors further explain what they mean by decorrelation because even if the authors published that result before this could help here to better understand.

To clarify, we have added the following text to the discussion, "In other words, odors that bind the most tightly to odor receptors do not always strongly activate the odor receptor complex to activate the downstream signaling cascade that ultimately generates OSN output." What we mean is that the binding affinity need NOT be correlated with activation efficacy.

L244: What do the authors mean by "plasticity" here, the presence of this word needs to be developed a little more or remove.

We consider the plasticity here to be a form of "short-term plasticity" whereby the stimuli (or mixtures thereof) influence the activity of receptor cells in non-linear ways. However, this may be a bit ambiguous, and we have changed the wording to "modulation" in our revision.

L244: through "chronic" the authors mean in vivo or during several sessions?

Here we mean both. Our approach allows for repeated imaging through the same window into the olfactory epithelium. Therefore, our approach will be valuable for potential future longitudinal experiments exploring OSN modulation.

L270-282: Even if they go beyond binary mixtures, may I suggest to the authors that this discussion about OSNs can comprise a little bit more citations of some literature dealing with antagonistic interactions with binary mixtures at OSN level. Especially the authors should refer to Chaput et al., 2012. As a matter of fact, the authors write “Antagonistic interactions have been hinted at previously, but we find direct evidence for widespread occurrence of this phenomenon.” I don’t fully agree with the verb “hint” Antagonistic interaction have been demonstrated though unit OSN recordings in vivo in rats, even if the in vivo results prior to their study remain poor in number.

By this we only mean that our approach verifies pervious work and also provides new evidence that the phenomenon is readily observed, rather than restricted to special cases between certain odors or ORN subtypes. In our revision we have cited the previous work (including Chaput et al 2012) and changed our language to include past observations of antagonism.

The new text reads, “Evidence of antagonistic interactions in OSNs has been reported previously and we advance these prior observations by providing evidence for widespread occurrence of this phenomenon.”

L278: Maybe the authors could discuss this fundamental result which was until now only a hypothesis in term of peripheral coding which increases in precision and specificity with odor stimulus complexity, finally here the system works in really natural condition by comparison with experiments which used monomolecular of binary mixtures. So, in physiological condition, the system gives proofs of a great specificity since the periphery.

We are glad that Dr. Viret agrees that our data shows strong evidence, under realistic conditions, for modulation of peripheral coding. We have added new text to the discussion to better highlight this point. Please see below:

“It has been previously hypothesized that such nonlinearities in peripheral coding may be a mechanism to increase precision and specificity of encoding complex odor stimuli (Chaput et al., 2012; Rospars et al., 2008; Reddy et al., 2018; Singh et al., 2019). Indeed, using more naturalistic odor delivery, we found that the degree of mixture suppression was greater with increasing number of components, which is expected from statistical consideration of antagonistic interactions¹¹.”

L289-292; “These additional data, along with our previous observation that narrowly tuned ORNs show strong mixture-suppression (Figure 291 5D), argue that antagonism is indeed a fundamental feature of mixture encoding in ORNs and is 292 not a unique feature restricted to those that are sensitive to a single functional group.”

Here, to my opinion, it is a crucial result that highly indicates that the very precise organization of the projection of OSNs onto glomeruli based on the OR expression ensures the maintenance of the hardware organization but cannot be directly and simply linked to a functional spatial coding; what needed to be demonstrated.

We are uncertain what Dr. Viret is conveying with this comment. If she means that anatomical and functional organization of the epithelium are not related – we agree. In fact, we find that antagonistic interactions are not specific to any one pair or odors or functional group specificity, but rather is a general feature of peripheral mixture encoding.

L283-289: This result that stimuli with close chemical features involve suppressive interactions was very important and could be discussed regarding the seminal results obtained in frog (group of Duchamp, Holley, Revial and Sicard in the seventies) with monomolecular stimuli which established qualitative space by comparing the unit qualitative profiles by pairs. What do you think about these results? Of course, they have been obtained in “physiological extreme conditions” since the system is not design to process monomolecular odorant but put in light of results obtained with complex blends, it could bring a new look on the qualitative groups established from monomolecular stimuli and further strengthen the authors’ contribution.

We are not entirely sure what Dr. Viret is referring to, but we think this comment is about how the neural representation (activity patterns of ORNs) varies for different pairs of odors – with an eye to classifying or grouping specific types of molecules or features. This sort of analysis, pioneered by the French physiologists Duchamp, Holley, Sicard and others (underappreciated in current times!), has been done in terms of chemotopic organization of glomeruli (eg, Uchida et al *Nat Neurosci* 3:1035; Johnson et al *J Comp Neurol* 409:529; Bozza et al *Neuron* 42:9; Soucy et al *Nat Neurosci* 12:210; Ma et al, *PNAS* 109:5481). While we agree that this topic is exciting and our data could speak to it, we feel that it is beyond the scope of the current paper, which already reports many different findings. We hope to follow up on Dr. Viret’s suggestion and look for interesting features in ORN responses – perhaps even in collaboration.

To give a taste for the sort of analysis we could do in the future, we performed hierarchical clustering of odors based on the ORN patterns. This analysis might inform us about any groupings of odors based on the similarity of their activation of ORNs. As seen in the dendrogram below, we see that many pairs of odors that we would anticipate to have similar representations (eg methyl and ethyl tiglate; + and – carvone; pentyl acetate and isoamyl acetate) do show up close to each other in this dendrogram. However, to do deeper and mechanistic analysis of this sort (and then relating it to antagonistic/suppressive interactions) will require additional experiments and analysis beyond the scope of this paper. We hope that Dr. Viret agrees.

L293-298: I fully agree with you. You should add: we report responses from a very large set of OSNs!

We have changed the text to read, “(1) we report responses from a large set of OSNs in intact, freely-breathing mice, responding to vapor phase odorants inhaled physiologically”

L300: I cannot see the widespread suppression as a normalization (but may I don’t well catch the meaning of the word), this word is negative for me. Of course, I agree with the authors the system must be preserve from saturation, but it is more than that. To my opinion the olfactory system makes exactly the contrary of the normalization, more complex is a stimulus, more compounds compose it, more precise et unique will be the OSN combination encoding it (moreover a plausible hypothesis is that fewer neurons are involved when the complexity increases) to my idea this does not match with the notion of normalization.

We respectfully contend that there may be a language misinterpretation in this comment. When we discuss “normalization,” we are referring to a very well described neural computation that is present in many sensory

systems. It is so prevalent that it is now referred to as a “canonical computational motif” (Carandini and Heeger, 2012, DOI: 10.1038/nrn3136; Louie and Glimcher, 2019, DOI: 10.1093/acrefore/9780190264086.013.43; Olsen et al 2010 DOI: 10.1016/j.neuron.2010.04.009). We have discussed this extensively in our theory paper (Reddy, Zak et al eLife 2018), which grounds this current paper. Despite this (presumed) misinterpretation, Dr. Viret does indeed correctly point out that the phenomenon we observe helps to prevent saturation of OSNs. To clarify, the formal name of this computation is “normalization”. As we noted above, the term normalization is NOT equivalent to making the activity uniform so as to lose discrimination (which may be Dr. Viret’s concern).

L328: To my view, it should be added that the surgical procedure allows to record the calcium activity of OSNs lying in the dorsal recess region of the OE. From the procedure description of the OB craniotomy I understand that the same mice can be recorded several times. Is it correct? On the other hand, for the OSN recordings, I guess that the authors make only one recording session? Is it correct? It should be better to clarify and to precise, if it is the case, that surgical procedure of the nose does not allow to keep the animal implanted to be chronically recorded.

It is correct that the same mice can be recorded several times using the OB craniotomy procedure. This is also true of the OSN recordings in the dorsal recess. In fact, because the bone is only thinned and not removed over the epithelium the stability of this preparation is superior to the OB craniotomy. We are able to image the same area of epithelium for > 6 months.

We have added the following text to the methods to clarify, “Following a recovery period, the thinned bone procedure allowed for chronic imaging of the epithelium for several weeks.”

L346: Odor stimulation: The best would be that the authors give a table gathering the odorants, such a table including CAS number of molecules, the saturating vapor pressure, molecular weight and concentrations used, the same applies to the mixtures, for which are needed the list of components, ratio and fraction of each and, in addition given that each stimulus has not the same volatility properties, the range over which each molecule was used. I think such information is important and fundamental to allow for future comparisons.

We now include a table of all odors used for each experiment as Dr. Viret suggests. Tables 1 & 2 contain information for the binary mixture experiments. Table 3 contains information for the odor tuning (Figure 5) and complex mixture experiments (Figure 7).

L351: “The absolute relative odor concentration was measured by a photoionization detector (PID; Aurora Scientific) and normalized to the largest detected signal for each odor” It is fine to make such measures and to make mixtures in air-phase. Please what does mean the absolute relative concentration? (the same question holds for the legend of Fig.1) Please what does mean “normalized to the largest signal for each odor?”

The signal on a PID is proportional to the concentration of an odor; however, it does not report the “real” concentration. For our study, we started with ~80% pure odor diluted in mineral oil (further diluted 16 times in air) and measured the resulting PID response. We could then determine the relative concentration of each of our other odor dilutions by comparing their PID response to the maximum PID signal. We “normalized” each dilution by dividing the PID signal by the signal for the highest odor concentration. We agree, however, that the term “absolute relative concentration” is confusing, and we simply call it relative concentration.

L355: It is very fine that the authors could check that quantitatively the mixture is the linear summation of each odorant alone. I would be perfect if the authors precise the chosen ratio of each compound; it can be done in the table suggested before.

For the binary mixture experiments we now include a table that provides the exact vapor pressure of each odor. For these experiments, the vapor pressure is closely matched between the two odors.

L357: Given the fundamental importance of the choice of odorant stimuli, can the authors say a little bit more about the choice of them regarding literature? Preceding experiments? Their effectiveness at the periphery? Etc.? Why did the authors select odorants from 1 to 16 for complex mixture stimuli?

Prompted by this comment, we have added the following new text to our revision: “Our choice of odors for this experiment was motivated by several previous studies that demonstrate their efficacy and diversity in activating OSNs that presumably signal through GPCR signaling pathways^{39–41}, as well as past behavioral work^{14,42}, which demonstrates that mixtures of these odors can be perceived and distinguished by mice despite their overlapping OSN activity patterns.”

L389: Can the authors describe more precisely the equation parameters of the response? the authors define RC, b and Rmax+k and k-1 which does not appear in the equation. May the authors clarify please.?

We have included new text in the methods to more completely describe our data fitting procedure and associated parameters. The new text is as follows:

“The four free parameters that are fit to the data are b , R_{max} , K and η . Here b is interpreted as the baseline response for blank odors, R_{max} is the response above baseline at saturating concentrations, K (commonly referred to as the dissociation constant) is the reciprocal of the binding affinity and η is the Hill coefficient. To account for noise and variability in the data, we used four criteria to select the fitted curves (all units are $\Delta F/F$): 1) The total squared deviation of the data from the best fit sigmoid shouldn't be too large. We used a threshold of 0.1 to ensure rejection of highly noisy data. 2) The maximum value over all concentrations should be greater than 0.4 $\Delta F/F$, 3) Binding affinity should be within the range of the concentrations used and 4) the hill coefficient cannot be below 0.75 or above 6 to exclude unreasonably sharp fits.

Fig5: Fig5C: the authors should clarify what do they mean by predictions, why only one box is blackened? Is it the best stimulus? Do the grey boxes mean that the responses to these stimuli were weaker? White boxes are for stimuli that did not elicit responses What are the individual odorants? Give the composition of mixture, components, ratio, PID measures if the authors had? Maybe all this information can be in the stimulus table.

We used the response of 16 individual odors (left) to make predictions about what mixture response would look like in individual OSNs. On the right, we made complex mixtures from a subset of the 16 individual odors. Black or grey mean that the odor was present in the mixture. White means that odor was absent. We left one box black (instead of making all present odors grey) to show that we could also hold one mixture component constant across many mixtures to look for interactions with other odors in the panel.

We do not have PID measurements for each of the mixtures (only the individual components, Supplementary Figure 5). The signal on the PID is actually quite noisy and requires averaging (5-10 trials) for to resolve a signal for many odors. Because we used 84 different mixtures, this would require >500 trials, which was time prohibitive. We have previously shown that mixtures in our olfactometer are faithful sums of the individual components (supplementary figure 1 in Rokni et al, *Nature Neuroscience* 17:1225).

For the composition of each mixture, please see new Supplementary Table 1.

Supplemental Fig2: This figure is really very interesting. Please can the authors precise the constitution of mixtures? Are there the same for the 5 OSNs?

Yes, each OSN for all complex mixture experiments was presented with the same 84 mixtures + 16 component odors. The composition of the mixtures is exactly the same as in Figure 5. Please see new Supplementary Table 1.

Supplemental Fig 5: A: PID measures are very useful, for example in A to compare the compound volatilities, we can observe here that Methyl tiglate and isobutyl prop. Are not very far from each other and that their mixture is really the summation of them.

We would also point out that the PID certainly indirectly measures compound volatilities, the primary measurement is the ionization energy of the odor molecule. Therefore, volatilities between two molecules with different ionization energies cannot be reliably compared. However, in this case it is fortunate that Methyl tiglate and isobutyl propionate have similar molecular shapes (ionization energy) and SVP (see Table 1).

C: What do the values to the left of the PID traces for the odorants correspond to i.e. to which concentration in % vol/vol, is it the same for each odorant?

We apologize for the oversight here. The scale bar is the voltage signal recorded on the PID for each odor. The scale is variable because the ionization energy for each odor varies and is influenced by its molecular shape. We have added new text to the legend to clarify. For each odor, the initial dilution was 16% v/v and a further air phase dilution 16 times for a final concentration of 1%. The true odor concentration is of course influenced by the vapor pressure which we now provide in Table 3 for each odor.

Reviewers' Comments:

Reviewer #1:

Remarks to the Author:

The addition of Figure S4 adequately addresses my concern. I actually prefer S4 to the figure 7 analysis, because it's simpler. Either way, I look forward to the publication of this important work.

Sincerely,

Matt Smear

Reviewer #2:

Remarks to the Author:

The authors have very nicely responded to my comments. I thank the reviewers for the new figure 4, and I accept their argument that the uncertainty about their calcium indicator dynamics precludes further analysis of response kinetics.

The authors have done good job addressing the comments of the other reviewers.

Reviewer #3:

Remarks to the Author:

I thank the authors for fully responding to all my requests and questions. In this version including, in particular, all the information about the stimuli and the additional figures, the work seems to me perfectly clear to be published. I hope that this publication will receive all the attention it deserves. I count on the authors to use their chronic freely-breathing mouse model to further advance on the mechanisms of peripheral encoding of complex mixtures (comparison of qualitative response profiles) and induced conditioning phenotypic peripheral plasticity (which is a current personal goal that I am exploring in the rabbit model). I also encourage them, if possible, to use real mixtures in future studies, i.e. including compounds at very different concentrations from each other.

Patricia Duchamp-Viret